# Molecular Profiling and Novel Therapeutic Strategies for Mucosal Melanoma: A Comprehensive Review

**DOI:** 10.3390/ijms23010147

**Published:** 2021-12-23

**Authors:** Alice Indini, Fausto Roila, Francesco Grossi, Daniela Massi, Mario Mandalà

**Affiliations:** 1Division of Medical Oncology, Department of Medicine and Surgery, Ospedale di Circolo e Fondazione Macchi, ASST dei Sette Laghi, 21100 Varese, Italy; alice.indini@gmail.com (A.I.); francesco.grossi@asst-settelaghi.it (F.G.); 2Unit of Medical Oncology, Department of Surgery and Medicine, University of Perugia, 06129 Perugia, Italy; fausto.roila@unipg.it; 3Faculty of Medicine and Surgery, University of Insubria, Ospedale di Circolo e Fondazione Macchi, 21100 Varese, Italy; 4Section of Pathological Anatomy, Department of Health Sciences, University of Florence, 50134 Florence, Italy; daniela.massi@unifi.it

**Keywords:** melanoma, mucosal, immunotherapy, targeted therapy, c-kit

## Abstract

Mucosal melanoma is a rare and aggressive subtype of melanoma. Unlike its cutaneous counterpart, mucosal melanoma has only gained limited benefit from novel treatment approaches due to the lack of actionable driver mutations and poor response to immunotherapy. Over the last years, whole-genome and exome sequencing techniques have led to increased knowledge on the molecular landscape of mucosal melanoma. Molecular studies have underlined noteworthy findings with potential therapeutic implications, including the presence of *KIT* mutations, which are potential targets of tyrosine kinase inhibitors currently in use in the clinic (imatinib), but also *SF3B1* mutation, *CDK4* amplifications, and *CDKN2A* gene deletions, which are presently under investigation in clinical trials. Recent results from a pooled analysis of patients with mucosal melanoma treated with immunotherapy have suggested that the combination of immune checkpoint inhibitors might improve survival outcomes in this subset of patients, as compared with single-agent immunotherapy. However, these results are not confirmed across different studies, and combo-immunotherapy correlates with a higher rate of adverse events. In this review, we describe the clinical, biological, and genetic features of mucosal melanoma. We also provide an update on the results of approved systemic treatment in this setting and overview the therapeutic strategies currently under investigation in clinical trials.

## 1. Introduction

Mucosal melanoma is a rare disease that is epidemiologically, biologically, and molecularly distinct from cutaneous melanoma [1]. Non-cutaneous melanocytes can be found in multiple sites of the human body, namely the ocular tract (including the uvea and the conjunctiva) and mucosal surfaces lining respiratory, gastrointestinal, and urogenital tracts [2]. With an overall incidence of 1.4 cases/million, mucosal melanomas represent approximately 1.5% of all melanoma cases and only 0.03% of all diagnosed cancers [3]. Mucosal melanomas most often arise in the head and neck cavities (55.4%), including the oral and nasal cavity and accessory sinuses, followed by the anus/rectum (23.8%) and the female genital tract (18%) (Figure 1) [3,4]. A variety of primary non-cutaneous melanoma has been described to occur in almost every part of the body, including the gastrointestinal tract, larynx, lungs, leptomeninges, or biliary tree [5,6]. However, in some cases, melanomas arising in atypical internal sites might represent metastases of occult primary cutaneous melanoma, as suggested by mutational analysis of a cohort of patients with presumed primary pulmonary melanomas, showing a high mutational load with ultraviolet (UV) signature and mutations in genetic drivers typical of cutaneous melanoma [7].

Compared to its cutaneous counterpart, mucosal melanoma represents a challenge for both diagnosis and treatment. Due to their hidden sites of origin, these tumors are often asymptomatic in their early stages and are usually diagnosed late. The presence of a thick vascularization on the site of disease onset allows early metastatic spread through the lymphatic and vascular networks. Altogether, these features explain the aggressive biology of mucosal melanomas, their overall poor prognosis, and low survival rates [1]. Moreover, the rarity of the disease makes large epidemiological studies and clinical trials difficult to be performed. Notwithstanding, over the last few years, the therapeutic perspective has started to change thanks to the growing knowledge on the molecular pathogenesis of mucosal melanoma and the use of novel treatment strategies. Still, most evidence suggests that the prognosis of patients with mucosal melanoma has not improved as much as for cutaneous melanoma [8]. In this review, we describe the clinical and biological features of mucosal melanoma, focusing on a small case presentation from our clinical experience. We also provide an update on the results of approved systemic treatment in this setting and overview the therapeutic strategies currently under investigation in clinical trials.

## 2. Materials and Methods

The papers referenced in this review were selected through a PubMed search performed on 1 November 2021, with the following searching terms: melanoma and mucosal, or non-cutaneous, or head and neck, genitourinary, or anal, or vulvovaginal. Oral presentation, abstracts, and posters presented at the American Society of Clinical Oncology (ASCO, Alexandria, VA, USA) 2021 and the European Society for Medical Oncology (ESMO, Lugano, Switzerland) 2021 annual meetings were retrieved for data on mucosal melanoma. We also overview ongoing research and data of combination therapies currently under investigation, which will impact future therapeutic strategies. Clinicaltrials.gov was searched to identify ongoing clinical trials enrolling patients with mucosal melanoma, both in the adjuvant and systemic settings.

Histologic and radiologic documentation were retrieved from patients’ clinical records in order to provide representative data upon patients’ (or their relatives’) written informed consent to use these images for research publication purposes (Figure 1 and Figure 2).

## 3. Genomic Profiling

Genetic alterations occurring in mucosal melanoma are different from those observed in cutaneous melanoma. Recently, a meta-analysis of major studies of whole genome and whole exome sequencing of mucosal melanoma has been published [9]. Compared to cutaneous forms, mucosal melanomas show lower rates of somatic mutations, do not display the UV mutational signatures and show increased somatic genomic instability [9]. The overall frequency of main mutations in mucosal melanomas is as follows (Table 1): *NRAS* (8%), *BRAF* (6%), neurofibromin 1 (*NF1*) (14%), *KIT* (13%), splicing factor 3b subunit 1(*SF3B1*) (15%) [9]. Other less commonly described mutations in mucosal melanomas are *TP53* (8.9%), sprout-related EVH1 domain containing protein 1 (*SPRED1*) (7%), ATP dependent helicase (*ATRX*) (6%), and Chromodomain Helicase DNA Binding Protein 8 (*CHD8*) (4%) [10,11,12]. However, the frequency of different mutations may vary across different studies and reports. Moreover, there are significant differences in the frequency of these mutations across specific anatomical sites of mucosal melanomas [9]. Head and neck melanomas, and specifically sinonasal melanomas, most commonly show mutations in *NRAS* and Telomerase Reverse Transcriptase (*TERT*) promoter, as compared with *BRAF* and *KIT* mutations [13,14,15]. Genitourinary tract and anorectal melanomas (both in males and females) frequently show *KIT* mutations and amplifications, together with *NRAS* mutations; notably, *KIT* mutations are uncommon in vaginal melanomas but are frequently present in vulvar melanomas [16]. Recurrent loss of *SPRED1*, a negative regulator of the mitogen-associated protein kinase (MAPK) pathway which acts by recruiting NF1 to the plasmatic membrane, has been found in 37% of 47 patients with mucosal melanomas, mostly of anorectal and vulvovaginal origin: this molecular alteration was mutually exclusive with *NF1* mutations, which were found in 12% of patients [17]. In a study utilizing a zebrafish model, Ablain et al. demonstrated an in vivo cooperation between loss of *SPRED1* and *KIT* activating mutations, suggesting that this molecular alteration might induce cell proliferation and confer resistance to KIT inhibitors [17].

Mutations in *BRAF* and *NRAS* are not so uncommon among mucosal melanomas (up to 20% of cases); rather, they have peculiar features that are radically different from those observed in cutaneous melanomas [18]. Mucosal melanomas show a lower frequency of *NRAS* Q61 mutations and a higher frequency of mutations in G12 and G13, suggesting that these are not linked to UV irradiation but possibly due to the effect of external genotoxic factors, which remain to be identified. Similarly, *BRAF* mutations in mucosal melanomas mainly consist of non-canonical non-V600 mutations and are similar to those observed in lung adenocarcinomas for both the location of mutated codons and the preferred amino acid substitutions (mostly prevalence of D594G, G469A, and K601E) [18]. Indeed, *BRAF* mutation was reported in 26% of patients with vulvar and vaginal melanoma in a case series of 51 patients; however, only 50% of them were V600 mutations [16]. *BRAF* fusions in mucosal melanomas have also been described, with a comparable frequency as of cutaneous triple wild type melanoma (i.e., lacking *BRAF*, *NRAS*, and *NF1* mutations) [19]. Interestingly, while being resistant to vemurafenib in vitro, the *ZNF767-BRAF* fusion showed in vitro and in vivo response to MEK inhibitor alone or combined with either Phosphatidylinositol-4,5-Bisphosphate 3-Kinase (PI3K) or cyclin-dependent kinases (CDK) 4/6 inhibitors [19]. The distribution of *BRAF* and *NRAS* mutation varies according to geographic regions, as observed in cutaneous melanoma, and might explain the better survival observed among European patients in comparison with patients from North America and Asia [20].

Loss of *NF1*, which acts as a negative regulator of Ras, is associated with increased MAPK pathway activity [21]. The rate of *NF1* mutations is comparable across cutaneous and mucosal melanomas [9]. Interestingly, the whole-exome sequencing study by Hintzsche et al. found that *NF1* was co-mutated with *KIT* in 32% of mucosal melanomas, which is a significantly higher rate if compared with cutaneous forms [22].

SF3B1 is a spliceosomal protein that plays a major role in RNA splicing. Thus, *SF3B1* mutations result in alternative splicing leading to an overall transcriptomic dysregulation [23]. *SF3B1* mutations, specifically the R625C and R625H, are the most commonly reported in uveal, vulvovaginal, and anorectal melanomas [22,24,25]. *SF3B1* mutation has been reported to be associated with late metastatization and better prognosis in patients with uveal melanoma [25]; however, it did not seem to have the same correlation in a recently published case series of mucosal melanomas [19]. Comparison of mutational profiles of upper versus lower body sites mucosal melanomas found *SF3B1* hotspot mutations in 27% of lower body (most commonly vulvar ad anorectal primary melanomas) compared to 6% in the upper body sites [9]. Conversely, nasal melanomas more frequently showed non-canonical SF3B1-E1105B mutations located in the heat domain of *SF3B1* [22].

Mutations in Insulin Like Growth Factor Receptor 2 (*IGF2R*) and Deleted in Colorectal Cancer (*DCC*) genes were reported to be strikingly more frequent among mucosal as compared with cutaneous melanomas, accounting for nearly 32% of patients in the study by Iida et al. According to the Tumor Cancer Genome Atlas (TCGA) database, such mutations are quite uncommon in other types of solid tumors, thus confirming the peculiar and unique genetic background of mucosal melanomas [12].

Anaplastic lymphoma kinase (*ALK*) fusions have been described in mucosal melanoma, specifically the *EML4-ALK* fusion and an alternate *ALK* isoform, *ALK-AT1*. However, such molecular alterations have no clear in vitro and in vivo sensitivity to ALK inhibition, and the therapeutic impact is yet to be clarified [26].

Recent studies of mucosal melanoma analyzed by next-generation sequencing demonstrated that mucosal melanomas have high chromosomal instability and often show structural variants of *CDK4*, *MDM2*, *TERT*, cyclin D1 (*CCND1*), and *NOTCH2*, leading to their amplification, along with losses of cyclin-dependent kinase inhibitor (*CDKN*) *2A/B*, phosphatase and tensin homolog (*PTEN*) and p53 [10,11,17]. Several tumor-intrinsic factors leading to immune escape and resistance to immune-checkpoint inhibitors have been characterized in cutaneous melanoma, including low tumor antigenicity, disruption of interferon-gamma (IFN-γ) signature, and loss of major histocompatibility complex (MHC) expression. Moreover, *PTEN* loss, amplification of MAPK and CDK4/6 pathways, and WNT-β catenin dysregulation are all oncogenic signals contributing to immunosuppression [27]. However, the role of such mechanisms in mucosal melanomas is still unknown [28]. Recurrent activating mutations in β-catenin gene (*CTNNB1*) were detected in mucosal melanomas of various primary sites, suggesting they might play a role in resistance to immunotherapy [10]. Moreover, the lower antigenic load, low tumor mutational burden, and programmed cell death ligand 1 (PD-L1) expression of non-cutaneous melanomas together with a higher level of aneuploidy might be associated with lower T cell activation and poor response to immunotherapy [12,29].

In conclusion, mucosal melanomas show a complex and peculiar genomic profile that contributes to their biology, aggressiveness, and accounts for the poor response to systemic therapies. Epigenetic mechanisms further complicate this picture, as they might influence response to treatment [30]. Several therapeutic strategies, mostly leveraged from previous experience of cutaneous melanoma studies, have been investigated in mucosal melanoma and will be detailed in the next section.

## 4. Therapeutic Approach

### 4.1. Targeted Therapy and Tyrosine Kinase Inhibitors

Given the low rate of BRAF V600 mutation among patients with mucosal melanomas, the combination of BRAF and MEK inhibitors has not led to the same clinical success observed in cutaneous melanoma harboring *BRAF* activating mutations. Still, there is preclinical evidence that *NF1* loss of function mutations or deletions might lead to increased resistance to BRAF inhibitors [31]. As such, *NF1* mutations together with *RAS* mutations and *BRAF* fusions, which have been described in mucosal melanomas and lead to increased activation of the MAPK pathway, are promising therapeutic targets for MEK inhibition.

Certain *KIT* alterations, namely the KIT exon 11 and 13 mutations, show response to KIT inhibitors such as imatinib, nilotinib, and dasatinib [32,33]. Figure 2 shows PET/CT scan images of a patient with KIT exon 11 mutated anorectal mucosal melanoma responding to first-line treatment with imatinib. Conversely, KIT exon 17 mutation, KIT amplification, or immunohistochemistry staining positivity on tumor tissue appear to have minimal or no sensitivity to KIT inhibitors. A recent meta-analysis of 19 studies of c-Kit inhibitors for unresectable or metastatic mucosal, acral, or chronically sun-damaged melanoma (overall sample size of 601 patients) showed a pooled objective response rate (ORR) of 14% for all inhibitors among patients with mucosal melanoma, with the highest ORR (20%) for nilotinib [34]. Serial tumor biopsies of patients treated with nilotinib in a French phase II clinical trial showed that patients with good response to nilotinib had persistently decreased levels of phospho-STAT3 as compared with poor responders to treatment [35]. This data underlines the potential role of phospho-STAT3 as a biomarker of response to nilotinib through increased activity of KIT inhibition through the downregulation of the downstream signaling protein STAT3.

The dysregulation of cell cycle progression, caused by *CDK4* amplification and/or *CCND1* amplification and/or p16 (*CDKN2A*) loss, is a key genetic feature in mucosal melanoma [11]. Targeting CDK4/6 is a promising therapeutic strategy that is supported by a strong preclinical rationale and some case reports of response to the CDK4/6 inhibitor, palbociclib [11,36,37]. To date, the use of CDK4/6 inhibitors in patients with mucosal melanoma harboring amplification of *CDK4* is under evaluation in clinical trials (Table 2).

SF3B1 mutation represents another promising target among patients with mucosal melanoma. Currently, the orally available spliceosomal inhibitor, H3B-8800, is under investigation in phase 1 clinical trials of patients with advanced myeloid malignancies harboring SF3B1-mutations [38]. Another therapeutic strategy that demonstrated antitumor activity in Asian patients with acral and mucosal melanoma after the failure of conventional treatment was the combination of an alkylating agent, temozolomide, with the vascular endothelial growth factor receptor 2 (VEGFR2) tyrosine kinase inhibitor, apatinib [39]. However, due to the limited efficacy observed in a single phase 1 study, this combination was not developed further.

### 4.2. Immunotherapy

The role of immune-checkpoint inhibitors (ICIs), including antibodies targeting the cytotoxic T lymphocyte 4 (CTLA-4), ipilimumab, and the programmed cell death 1 (PD-1), nivolumab, and pembrolizumab, as monotherapies in mucosal melanoma, has been assessed in several monocentric and multicentric retrospective series [40,41,42,43]. Overall, ORR and progression-free survival (PFS) rates have been found to be comparable, but most often worse, compared with cutaneous melanomas, regardless of the primary site of origin of mucosal melanomas. A post-hoc analysis of three trials using pembrolizumab in metastatic melanoma, the KEYNOTE-001, KEYNOTE-002, and KEYNOTE-006, showed an ORR of 19% (95% CI, 11–29%), median PFS of 2.8 months (95% CI, 2.7–2.8), and the median OS was 11.3 months (95% CI, 7.7–16.6) among 84 patients with mucosal melanoma out of 1567 total patients enrolled [44]. Nivolumab monotherapy in patients with rare melanoma subtypes who progressed on or after ipilimumab treatment was evaluated in the phase II CheckMate 172 study [45]. This trial enrolled 1008 patients, of whom 6.3% (*n* = 63) had mucosal melanoma. Results from this trial showed that both mucosal and ocular melanomas had lower median overall survival (OS) compared with acral cutaneous and non-cutaneous melanoma, being median OS 11.5 months (95% CI, 6.4–15.0) and 18 month OS rates 31.5% for patients with mucosal melanoma [45].

In a pooled analysis of studies using nivolumab monotherapy, including 86 patients with mucosal melanoma, median PFS was 3.0 months (95% CI, 2.2 to 5.4 months) and 6.2 months (95% CI, 5.1 to 7.5 months) for mucosal and cutaneous melanoma, with objective response rates of 23.3% (95% CI, 14.8% to 33.6%) and 40.9% (95% CI, 37.1% to 44.7%), respectively [46]. The combination of nivolumab and ipilimumab has yielded more promising results even among patients with mucosal melanomas, with improved outcomes compared with both agents used as monotherapies [47,48]. In the pooled analysis of nivolumab studies, median PFS in patients treated with combined nivolumab and ipilimumab was 5.9 months (95% CI, 2.8 months to not reached) and 11.7 months (95% CI, 8.9 to 16.7 months) for mucosal and cutaneous melanoma, and the ORR was 37.1% (95% CI, 21.5% to 55.1%) and 60.4% (95% CI, 54.9% to 65.8%), respectively [46]. The 5-year follow-up of 79 patients with mucosal melanoma from the CheckMate-067 trial showed that the combination of ipilimumab and nivolumab resulted in considerable higher ORR compared with ipilimumab alone (43% vs. 7%), but also in higher complete response rate (14% vs. 0%), and OS rate (36% vs. 7%) [49].

Altogether, these trials included patients with different baseline characteristics, thus making it difficult to compare results. However, evidence to date has supported the role of anti-PD1 either as monotherapy or combined with ipilimumab, as a first-line therapeutic strategy for the treatment of patients with advanced unresectable/advanced mucosal melanoma. In this setting, multimodal treatment strategies combining radiotherapy with immunotherapy might lead to a boosted immune response with increased antitumor efficacy and improved local tumor control and symptoms relief [50,51,52].

Notably, clinical outcomes with ICIs seem to be poorer for Asian patients as compared with Caucasian patients with mucosal melanoma. In the KEYNOTE-151 trial, pembrolizumab provided an ORR of 13.3% in 15 Chinese patients with advanced/metastatic mucosal melanoma [53]. Similarly, the POLARIS-01 study showed poor clinical outcomes of toripalimab, a humanized antibody targeting PD1, among 22 Chinese patients with mucosal melanoma, with an ORR of 0%, a median OS of 10.3 months, and a median PFS of 1.9 months [54]. Differences in the biologic profile, namely a higher prevalence of mutations affecting the WNT-β catenin pathway and the high frequency of *KIT* mutations in an Asian population, might be responsible for the poor responses observed during treatment with ICIs [9,55,56,57].

## 5. Ongoing Clinical Trials and Future Perspectives

Due to the limited efficacy of single-agent targeted and immunotherapy in mucosal melanoma, several combination strategies are currently under investigation in clinical trials (Table 2). Combination therapies aim at targeting multiple mechanisms by which the cancer cell proliferate and evade the immune surveillance, thus exerting a synergistic antitumor effect. Promising response rates and survival results have been reported with the combination of nivolumab and ipilimumab, as detailed in the previous section.

VEGF expression was associated with poor survival outcomes among patients with primary mucosal melanoma of the oral cavity, suggesting it could be a potential therapeutic target. Unfortunately, preliminary data of antiangiogenic drugs, either alone or combined with cytotoxic chemotherapy, have yielded only limited antitumor activity and therapeutic results [37,58]. Inhibition of the VEGF signaling pathway also exerts immune-mediated effects in the tumor microenvironment. The combination of VEGFR inhibitors and anti-PD1 antibodies has also been investigated among patients with mucosal melanoma, showing promising results. The randomized phase III trial LEAP-003 of pembrolizumab combined with lenvatinib for first-line treatment of metastatic melanoma also included patients with acral and mucosal melanoma and is currently ongoing (NCT03820986). Bevacizumab combined with the anti-PD-L1 antibody, atezolizumab, has been investigated in a phase II trial (NCT04091217). A phase 2 trial of the anti-PD1 antibody, camrelizumab in combination with the multitargeted tyrosine kinase inhibitor, anlotinib, and nab-paclitaxel as first-line treatment of mucosal melanoma is currently ongoing and recruiting patients (NCT04979585).

The combination of toripalimab and axitinib showed preliminary clinical activity in a population of 29 treatment-naïve Asian patients with metastatic mucosal melanoma in a phase IB clinical trial [59]. The ORR was 48.3% (95% CI, 29.4–67.5%), and the median PFS was 7.5 months (95% CI, 3.7-not reached). In this trial, PD-L1 expression and high tumor mutational burden (TMB) were associated with higher ORR and better PFS. Given the above-mentioned differences in prognosis and response to immunotherapy between Asian and non-Asian patients, this combination therapy should be validated in a randomized phase III trial, also including a non-Asian population, in order to confirm its activity.

Preliminary data from a phase II study showed promising activity of the combination of toripalimab with the multi-target tyrosine kinase inhibitor vorolanib (CM082). In this study (*n* = 38 patients evaluable for response), the ORR was 22.2%, the disease control rate as 55.5%, and median PFS 5.7 months (NCT03602547) [60].

Novel agents currently under evaluation in clinical trials for mucosal melanoma include the CDK inhibitor, dinaciblib (NCT00937937); the MDM2 inhibitor, APG-115, in combination with pembrolizumab (NCT03611868); the engineered interleukin 2, nemvaleukin alfa as a single agent (NCT04830124), and aldesleukin in combination with pembrolizumab (NCT02748564). Several trials are also evaluating the role of multimodal treatment integration (i.e., radiotherapy and radiosurgery) with immunotherapy in order to improve therapeutic results. Moreover, trials are underway in order to assess the efficacy of immunotherapy, either as single-agent anti-PD1 or as a combination of anti-PD1 and anti-CTLA4 as neoadjuvant and/or adjuvant treatments for locally advanced high-risk mucosal melanoma before and after surgery.

## 6. Conclusions

Mucosal melanoma is an aggressive disease with an overall poor prognosis. To date, only limited therapeutic benefits have been obtained with KIT inhibitors and immunotherapy (either as single-agent anti-PD1 or as a combination of anti-PD1 and anti-CTLA4). However, response rates and survival outcomes are still worse than those observed among patients with cutaneous melanoma. Increasing knowledge of the genetic profile of mucosal melanoma by means of whole genome and whole exome sequencing has allowed a better understanding of the biology of this disease and has paved the way for potential therapeutic targets. Further prospective studies are of utmost relevance in this setting, where standard therapies are less efficacious than in cutaneous melanoma. At the present time, several clinical trials are ongoing to assess the role of systemic treatment in the perioperative setting for potentially resectable disease and in the adjuvant setting after radical surgery for localized disease. Moreover, several combination strategies for advanced/metastatic or relapsed disease are under evaluation in clinical trials, mainly consisting in the combination of immunotherapy with radiotherapy or antiangiogenic drugs, but also with novel emerging compounds targeting other signaling pathways. Cooperative data collection in order to identify clinical factors that influence disease response to treatment will help us gain more insight and eventually improve outcomes of this rare disease.

## Figures and Tables

**Figure 1 ijms-23-00147-f001:**
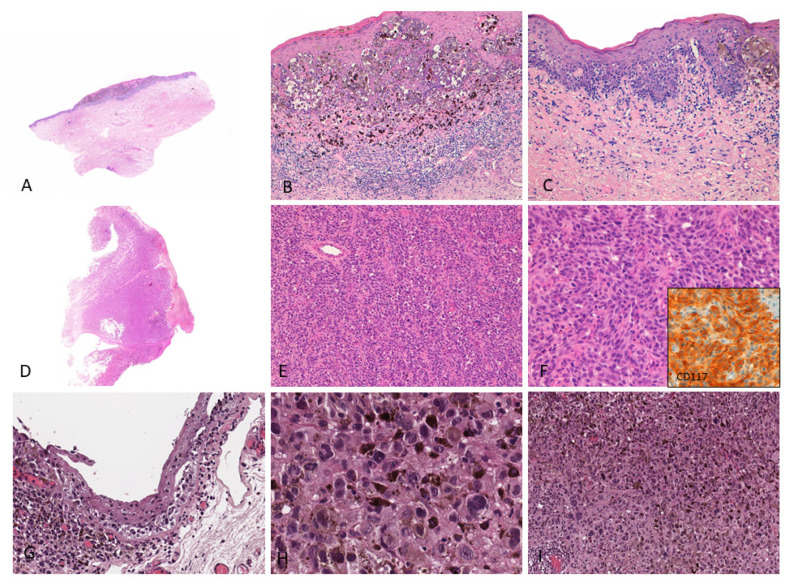
(**A**–**C**) Female, 76 years–Mucosal melanoma on the vulva (labia minora) characterized by atypical pigmented melanocytes in single units and nests with pagetoid spread; a brisk lymphocytic infiltrate with numerous melanophages is observed in the upper dermis; (**D**–**F**) Male, 86 years–Endoscopic biopsy reveals a bulky tumor diagnosed as amelanotic mucosal melanoma of the anal canal; the tumor shows diffuse proliferation of non-pigmented epithelioid cells; tumor cells show cytoplasmic and membranous CD117 staining (inset); (**G**–**I**) Female, 47 years–Highly pigmented conjunctival mucosal melanoma (left eye) with prominent melanin pigment in intracellular and extracellular location; both in situ (**G**) and invasive components (**H**,**I**) are shown. These images come from the pathology archive of patients with mucosal melanomas, as indicated in the “Materials and methods” section.

**Figure 2 ijms-23-00147-f002:**
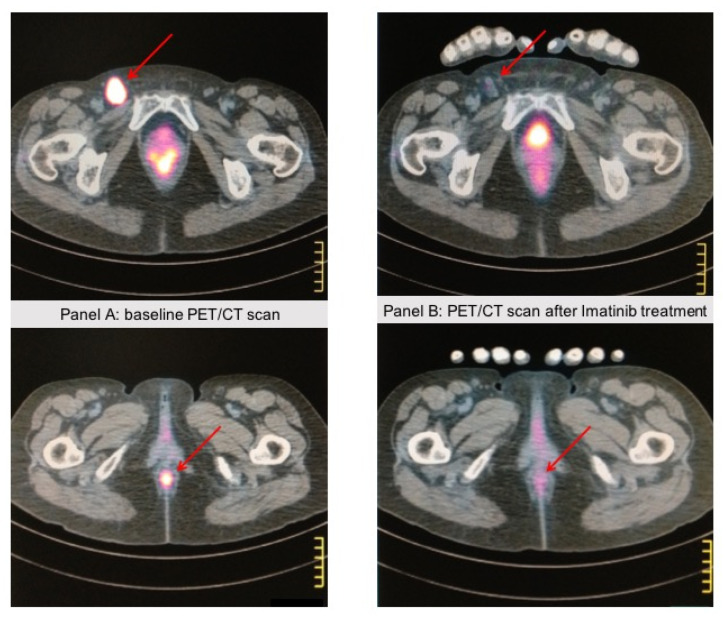
A 67-year-old patient came to our attention because of rectorrhagia; a PET/CT scan showed lung and abdominal lymph node metastases. A biopsy of the anal mass showed atypical epithelioid cells with pleomorphic features. Immunohistochemistry showed strong positivity for HMB45 and Melan A, whereas cytokeratins and S-100 expression were negative, consistent with the diagnosis of anal mucosal melanoma. c-KIT staining was positive in >75% of tumor cells, and DNA sequence analysis revealed a *KIT*-activating mutation (L576P) in exon 11. The patient received Imatinib 400 mg/die. Panel (**A**) shows the anal baseline lesion with a pathologic left inguinal lymph node. Panel (**B**) shows the PET/CT scan after 4 months of Imatinib treatment, showing a near-complete response in the anal lesion and the inguinal lymph node. The response lasted 3 years, then the patient progressed and died. These images come from the radiology archive of a patient with mucosal melanomas, as indicated in the “Materials and methods” section.

**Table 1 ijms-23-00147-t001:** Summary of most common genetic alterations in mucosal melanoma [9,13,14,15,16,17,18,19,22,23,24,25,27,28].

Gene	Molecular Alteration (s)	Frequency	Distribution across Melanoma Subtypes	Therapeutic Implications
BRAF	V600	6%	similar rates in upper and lower body regions	MAPK inhibition (BRAF and MEK inhibitors)
	D594G G469A, G469A K601E L505H, L597R T599I	5–20%	N.A.	Unknown response to MAPK inhibition
	ZNF767 fusion	N.A.		MEK inhibitor +/− PI3K or CDK4/6 inhibitors
NRAS	Q61 G12 G13	8–10%	43% vaginal melanomas 37.5% esophageal melanomas	Potential role of MEK inhibitors
KIT	Amplifications and missense mutations (Ex11 and Ex12-21)	13%	similar rates in upper and lower body regions	Imatinib Dasatinib Nilotinib
NF1	Loss of function mutations	14%	10% upper body regions 18% lower body regions	Potential role of MEK inhibitors
SF3B1	R625	15%	6% upper body regions 27% lower body regions	H3B-8800 (clinical trials)
SPRED1	Loss of function mutations	37%	most common among anorectal and vulvovaginal melanomas	No actual clinical applications Potential resistance to KIT inhibitors
TERT CDK4	Co-amplification	N.A.	>50% oral melanomas	Potential role of CDK4/6 inhibitors

Abbreviations: CDK4/6: cyclin-dependent kinase 4/6; MAPK: mitogen-associated protein kinase; N.A.: not assessed; PI3K: Phosphatidylinositol-4,5-Bisphosphate 3-Kinase.

**Table 2 ijms-23-00147-t002:** Overview of the main ongoing clinical trials (i.e., recruiting and active, not recruiting) for patients with mucosal melanoma in the adjuvant and metastatic setting (source: clinicaltrials.gov; and rctportal.niph.go.jp (accessed on 9 November 2021)).

Trial Name, NCT Number	Type of Study	Condition (s)	Drug (s)	Estimated Sample Size	Primary Endpoint (s)
**Perioperative (including neoadjuvant and adjuvant)**					
NCT03178123	Phase 2, randomized	Mucosal melanoma that has been removed by surgery	toripalimab high-dose recombinant IFN-a2b	*n* = 220	RFS (time frame: 5 years)
NCT04180995	Phase 2, single-arm	Localized mucosal melanoma considered to be able to be completely resected	neoadjuvant toripalimab + axitinib (8 weeks); adjuvant toripalimab (up to one year)	*n* = 30	Pathological response rate (pCR and pPR)
SALVO, NCT03241186	Phase 2, single-arm	Mucosal melanoma that has been removed by surgery (R0 or R1)	cycles 1–4: ipilimumab 1 mg/kg + nivolumab 3 mg/kg q3w; cycles 5–15: nivolumab 480 mg q28d	*n* = 30	RFS
IMMUQ, NCT03313206	Phase 2, single-arm	Resectable head and neck mucosal melanoma, amenable of post-operative RT	neoadjuvant pembrolizumab 200 mg q3w (up to 4 doses); adjuvant pembrolizumab 200 mg q3w (up to one year)	*n* = 50	DFS
NCT04622566	Phase 2, single-arm	Resectable mucosal melanoma	neoadjuvant pembrolizumab 200 mg q3w + lenvatinib QD (6 weeks); adjuvant pembrolizumab 200 mg q3w (up to one year)	*n* = 26	pCR rate
NCT05111574	Phase 2, randomized	Mucosal melanoma that has been removed by surgery	nivolumab + cabozantinib/placebo	*n* = 99	RFS
MEL60, NCT02126579	Phase 1/2, randomized	Resected stage IIB/IV melanoma	LPV7 + TLR agonists	*n* = 62	Incidence of AEs, T cell response in peripheral blood
NCT04879654	Phase 2, single-arm	Sinonasal melanoma removed with endoscopic surgery	toripalimab + CT + RT	*n* = 45	OS
NCT02519322	Phase 2, randomized	Stage III or oligometastatic stage IV that can be removed by surgery	Arm A: neoadjuvant nivolumab q2w for 4 doses; adjuvant nivolumab q2w for 13 doses Arm B: neoadjuvant nivolumab + ipilimumab q3w for 3 doses; adjuvant nivolumab q2w for 13 doses Arm C: neoadjuvant nivolumab + relatlimab q28d for 2 doses, adjuvant nivolumab + relatlimab q28d for 10 doses	*n* = 53	Proportion of patients with pathologic response to neoadjuvant nivolumab and ipilimumab plus nivolumab therapy
NCT03698019	Phase 2, randomized	Stage IIIB/C-IV resectable high-risk melanoma	Adjuvant pembrolizumab q3w for 18 cycles vs. neoadjuvant pembrolizumab q3w for 3 cycles, followed by adjuvant pembrolizumab q3w for 15 cycles	*n* = 500	EFS
**Metastatic**					
MTAM, NCT04472806	Phase 2, single-arm	Unresectable locally advanced/metastatic mucosal melanoma	toripalimab + endostar + CT	*n* = 31	PFS
NCT04318717	Phase 1/2, single-arm	Mucosal melanoma of the head and neck	pembrolizumab + hypofractionated RT	*n* = 16	Local tumor control rate
ARTISTRY-6, NCT04830124	Phase 2, single-arm	Mucosal and cutaneous melanoma who have progressed on previous anti-PD1/PD-L1 therapy	nemvaleukin alfa (SC or IV)	*n* = 110	ORR
BJCH-MM-0624, NCT03941795	Phase 2, randomized	Unresectable locally advanced/metastatic mucosal melanoma	toripalimab + axitinib vs. toripalimab vs. axitinib	*n* = 99	PFS
NCT03986515	Phase 2, single-arm	Advanced mucosal melanoma who have progressed after CT	apatinib + camrelizumab	*n* = 40	ORR
NCT04091217	Phase 2, single-arm	Unresectable locally advanced/metastatic mucosal melanoma	atezolizumab + bevacizumab	*n* = 43	ORR
NCT02978442	Phase 2, single-arm	Unresectable locally advanced/metastatic mucosal or acral lentiginous melanoma	cycles 1–4: ipilimumab 1 mg/kg + nivolumab 3 mg/kg q3w; followed by nivolumab 3 mg/kg q2w	*n* = 14	ORR
NCT04979585	Phase 2, single-arm	Unresectable locally advanced/metastatic mucosal melanoma	anlotinib + camrelizumab + nab-paclitaxel	*n* = 66	ORR
NCT05089370	Phase 1b/2	Unresectable locally advanced/metastatic mucosal melanoma	decitabine/cedazuridine + nivolumab	*n* = 30	Safety
PN21-001, NCT05098210	Phase 1	PD-1 inhibitor-refractory stage IIIC/IV melanoma	personalized multi-peptide Neo-Antigen Vaccine	*n* = 20	Incidence of AEs
NCT028748564	Phase 1b/2	Unresectable locally advanced/metastatic mucosal melanoma	aldesleukin + pembrolizumab	*n* = 65	ORR OTD
NCT03611868	Phase 1b/2	PD-1/PD-L1 inhibitor-refractory/relapsed melanoma	APG-115 + pembrolizumab	*n* = 203	MTD, RP2D ORR
NCT03025256	Phase 1/1b	Melanoma with leptomeningeal disease	intravenous + intrathecal nivolumab	*n* = 50	AEs, RP2D OS
NCT03865212	Phase 1	Metastatic melanoma	recombinant VSV-expressing IFN-beta and TYRP1	*n* = 72	AEs, MTD
NCT02535078	Phase 1b/2	Metastatic melanoma	IMC-gp100 + durvalumab or tremelimumab	*n* = 317	DLTs ORR
NCT04653038	Phase 1	Unresectable locally advanced/metastatic mucosal melanoma	MGD013	*n* = 160	ORR

Abbreviations: AEs: adverse events; CT: chemotherapy; DFS: disease-free survival; DLT: dose-limiting toxicity; EFS: event-free survival; IFN: interferon; IV: intravenous; LPV7: long peptide vaccine 7; MTD: maximum tolerated dose; ORR: objective response rate; OS: overall survival; OTD: optimal tolerated dose; pCR: pathologic complete response; PFS: progression-free survival; pPR: pathologic partial response; q28d: once every 28 days; q3w: once every three weeks; RP2D: recommended phase 2 dose; RT: radiotherapy; SC: subcutaneous; TLR: toll-like receptor; TYRP-1: tyrosinase related protein 1; VSV: Vesicular Stomatitis Virus.

## Data Availability

No new data were created or analyzed in this study. Data sharing is not applicable to this article.

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
