# Peer review of "Molecular Profiling and Novel Therapeutic Strategies for Mucosal Melanoma: A Comprehensive Review"

_ijms, 2021, doi:10.3390/ijms23010147_

Round 1
Reviewer 1 Report
The study demonstrates the genetic alterations in mucosal melanoma and the therapeutic implications.
- Abstract may be revised to indicate the tyrosine kinase inhibitors currently used in clinic for the treatment of mucosal melanoma.
- Figure 2 needs to be revised to show the labels for all four images.
Author Response
The study demonstrates the genetic alterations in mucosal melanoma and the therapeutic implications.
Comment: Abstract may be revised to indicate the tyrosine kinase inhibitors currently used in clinic for the treatment of mucosal melanoma.
Response: In order to comply with the reviewer’s suggestion, we indicated the tyrosine kinase inhibitor currently in use in the clinic for the treatment of mucosal melanoma.
Comment: Figure 2 needs to be revised to show the labels for all four images.
Response: Please note that the label “Panel A” is for the left-sided images, showing the anal baseline lesion with a pathologic left inguinal lymph-node; while, the label “Panel B” is for the right-sided images, and it shows the PET/CT scan after 4 months of Imatinib treatment.
Reviewer 2 Report
This review is well-written and appropriately structured. However, the manuscript requires spelling/grammar check and addition/removal of certain discussion points.
Specific comments:
- There are some grammar and typo mistakes that require corrections. For example, line 315 "fist-line treatment" should be "first" and not "fist".
- Section 2, "Materials and methods", is inappropriate in a review paper and should be removed.
- In the section of "Genomic profiling", the review lacks of the findings generated from gene expression profiling or RNA sequencing. Gene signatures are important in identifying MM patients who can response to immunotherapy. For example, the researchers from CU Center for Rare Melanomas found that mucosal melanomas had a much lower expression of a key critical innate immune-sensing pathway, making them less likely to respond to standard immunotherapy. Can the authors add those work in this review paper?
Author Response
This review is well-written and appropriately structured. However, the manuscript requires spelling/grammar check and addition/removal of certain discussion points.
Comment: There are some grammar and typo mistakes that require corrections. For example, line 315 "fist-line treatment" should be "first" and not "fist".
Response: We thank the reviewer for this correction, and checked all the manuscript for typos and misspelled words.
Comment: Section 2, "Materials and methods", is inappropriate in a review paper and should be removed.
Response: The IJMS Editorial Office specifically required a “Materials and methods” to be included in the review, to indicate the source of radiologic and histologic documentation, and the permission to use these data for research publication purposes.
Comment: In the section of "Genomic profiling", the review lacks of the findings generated from gene expression profiling or RNA sequencing. Gene signatures are important in identifying MM patients who can response to immunotherapy. For example, the researchers from CU Center for Rare Melanomas found that mucosal melanomas had a much lower expression of a key critical innate immune-sensing pathway, making them less likely to respond to standard immunotherapy. Can the authors add those work in this review paper?
Response: We discussed the role of next generation sequencing for mucosal melanoma and its role in determining response/resistance to immunotherapy in the “Genomic profiling” section (Lines 147-163). We acknowledge that there are studies ongoing on RNA sequencing looking at gene expression in mucosal melanoma. However, to the best of our knowledge these studies are presently underway and no data have been presented yet in the Literature.
Round 2
Reviewer 1 Report
The authors revised the manuscript according to the comments.